# A Low to Medium-Shear Extruded Kibble with Greater Resistant Starch Increased Fecal Oligosaccharides, Butyric Acid, and Other Saccharolytic Fermentation By-Products in Dogs

**DOI:** 10.3390/microorganisms9112293

**Published:** 2021-11-04

**Authors:** Isabella Corsato Alvarenga, Matthew I. Jackson, Dennis E. Jewell, Charles G. Aldrich

**Affiliations:** 1Department of Grain Science & Industry, Kansas State University, Manhattan, KS 66506, USA; icorsato@colostate.edu (I.C.A.); djewell@ksu.edu (D.E.J.); 2Pet Nutrition Center, Hill’s Pet Nutrition Inc., Topeka, KS 66617, USA; matthew_jackson@hillspet.com

**Keywords:** dog, microbiota, resistant starch, extrusion, corn, metabolomics

## Abstract

The objective of this study was to assess whether diets with increased resistant starch (RS) had a positive effect on markers of colonic health in dogs. Three identical diets were extruded with high, medium and low shear (HS, MS and LS) to incrementally increase RS, and fed to 24 dogs in a replicated 3 × 3 William’s Latin square design for 28-day periods. Fasting blood and fresh feces were collected on the last week of each period. Fecal quality was maintained among treatments. Gut integrity markers were measured by ELISA. Fecal short-chain fatty acids (SCFAs) were measured by LC MS/MS. In addition, the microbiota of dogs was determined from fresh feces by 16s rRNA high throughput sequencing. Untargeted metabolomics of both feces and serum were determined by UPLC. Data were analyzed using mixed models. There were no treatment effects on satiety hormones or gut integrity markers. Dogs fed LS or MS diets had marginal evidence (*p* < 0.10) for decreased fecal pH and for higher concentration (*p* < 0.05) of butyric acid and fecal oligosaccharides, succinate and lactate. Also, dogs fed the MS or LS diets had a shift towards more saccharolytic bacteria.

## 1. Introduction

Pet food companies are in constant search of innovation for additional health benefits for their commercial products. This is frequently accomplished by exploring novel or exotic ingredients in their dietary formulae or by modifying ingredients already present in the food. Such modifications could affect nutrient availability in commonly used ingredients, such as starches in grains. Starch ingredients that are extruded with low mechanical energy retain starch in a form that has been found to be resistant to digestion) both in vitro [1] and in vivo [2,3,4]. Resistant starches (RS) escape small intestinal digestion and absorption, and reach the colon, where their prebiotic activity may provide health benefits [2,3,4]. To date, most extruded pet foods produced in the US have little to no RS due to extensive starch gelatinization caused by the thermomechanical process [5]. Previous studies have explored health implications for dogs fed a low and high shear extrusion processes focused on local colonic effects [2,3,4]. 

The objective of this study was to assess whether diets with increased RS had a positive effect on the colonic health of dogs. We explored systemic and local endpoints of colonic health in dogs consuming diets extruded at three levels of mechanical energy and characterized microbiota changes in the colon related to consumption of these foods. The study hypothesis was that diets produced with less thermomechanical energy (i.e., low shear; LS) would promote an increase in short-chain fatty acids (SCFA) production in the colon of dogs, as well as increased satiety, fecal IgA (i.e., improved local immunity), and decreased serum lipoteichoic acid (LTA) and lipopolysaccharide (LPS) (i.e., decreased endotoxemia), relative to the low RS treatment (HS). Due to more extensive fermentation, we further hypothesized a decrease in colonic pH and an increase in lactate in dogs fed the LS treatment compared to the HS diet. The medium shear (MS) treatment was anticipated to present intermediate effects of both extremes.

## 2. Materials and Methods

### 2.1. Dietary Treatments

A single diet formulation was generated which met the maintenance nutrient requirements of adult dogs [5], with whole yellow corn included at 65% as the sole starch source. The remainder of the formula included 20% chicken meal, 8.4% pork fat, as well as minerals, vitamins and palatants. This ingredient blend was extruded at a target extruder shaft speed of 460, 375 and 250 rpm and increased in-barrel moisture as shear decreased [6]. Dietary treatments were thus defined as high, medium, and low shear foods (HS, MS, and LS, respectively), resulting in lowest, intermediate and largest concentrations of RS, respectively. For each dietary treatment, nutrients were analyzed at a commercial laboratory (Eurofins; Des Moines, IA, USA; Appendix A), according to their respective analytical methods (Appendix A). 

### 2.2. Preliminary Assessment of Digestibility & Palatability

For each dietary treatment, digestibility was estimated prior to the gut health study with three independent tests with six dogs each. This was a preliminary assessment for diet suitability for the gut health study presented herein. Each of the three dietary treatment apparent total tract digestibility (ATTD) trials used six Beagle dogs over 10 days per diet, with 5 d adaptation and total feces collected during the last 5 days, according to official procedures [5]. Samples of fresh feces were collected and frozen at −80 °C until analysis. Proximate analyses were conducted on diets and on feces and included dry matter (100%—moisture %; AOAC 930.15), ash (AOAC 942.05), organic matter (100%—moisture%—ash%), gross energy (measured as total heat of sample combustion by calorimetry), crude protein (AOAC 2001.11), true protein (feces only; crude protein in feces—estimated endogenous losses in feces), crude fat by acid hydrolysis (AOAC 954.02), and crude fiber (AOAC 962.09). Digestibility coefficients were calculated as (nutrient intake—nutrient fecal output)/ nutrient intake, on a dry matter basis. The ash ATTD in one dog consuming the HS diet presented a negative value; this was considered a technical error and the observation was excluded from further consideration. True protein digestibility was calculated as (protein intake—endogenous protein losses—fecal protein excreted)/protein intake; where endogenous protein losses were estimated as 394 mg/kg BW(kg)^0.75^/day [7,8]. While crude protein accounts for all the protein measured in the feces, true protein considers the endogenous losses in the feces from host protein and microbial mass.

Fecal quality of dogs fed dietary treatments during the preliminary digestibility study was determined according to a subjective scale from 1 to 5, recorded in 1 point increments; wherein, 1 = liquid feces (diarrhea), 2 = very soft feces, with little shape retention; 3= greater than 75% soft feces, less than 25% liquid feces, with some shape retention; 4= greater than 75% shape retention, with possible segmentation, and between 50–80% firm feces; and 5 = retained shape, above 80% firm feces, without segmentation. On this scale, scores above 3 were considered acceptable. Diet palatability was determined in 25 dogs with a two-bowl forced choice test over two days with bowl position switched for each day to remove side-bias. Intake ratio of each animal was determined as the ratio of total consumption of food A as a ratio of combined total consumption of foods A and B.

### 2.3. Gut Health Study

#### 2.3.1. Animals and Study Design

Eighteen adult healthy Beagle dogs and 6 adult mixed breed dogs were recruited for the feeding study at the Hill’s Pet Nutrition Center kennel (Topeka, KS, USA). The experimental design was that of a replicated Williams Latin square design, whereby each dog received all three dietary treatments over three periods. Each dog was assigned to one of six feeding sequences arranged in two squares balanced for carry over effects, with four dogs assigned to each sequence. Participating dogs consisted of two intact females, 9 neutered males, and 13 spayed females, with an average age (±standard deviation) of 5.1 years ± 3.20 (range 1.16 to 12.44 years old), and average BW (standard deviation) of 10.6 kg ± 2.26 (range 6.8 to 15.8 kg). Dogs were weighed weekly and food offerings adjusted to maintain weight. The study was approved by the Institutional Animal Care and Use Committee (IACUC) at Hill’s Pet Nutrition (protocol number 883.0.0.0).

Dogs were housed in one of three neighboring buildings and fed individually with an automated feeding system once (at 07:30) each day; fresh water was available at all times. Although dogs were caged in pairs, only one dog of each pair was included in the study. Dogs were allowed daily socializing time in a common playground area as part of the enrichment program. Before study initiation, dogs were fed a common baseline adult maintenance diet for 28 days. Dietary treatments were fed over three periods of 28 d each with no washout time in between periods.

#### 2.3.2. Sample Collection and Handling

Blood and fresh feces were collected at the beginning of the study (day 0), and towards the end of each 28-d treatment period. Specifically, fresh feces were collected within 15 min of defecation and within one hour after feeding (07:30 to 08:30) from day 24 to 26 of each period. Blood was collected on days 27 and 28 of each period. Dogs were not allowed access to the outside area 2 days before and during the collection week in order to avoid ingestion of grass. One fresh fecal sample was collected per dog per period and scored for consistency based on a subjective scale described above. Next, feces were homogenized (model ARM-310; Thinky Corporation, Tokyo, Japan) with planetary mixing at 2000 rpm for 30 s. Then, aliquots of fresh homogeneous feces were dispensed into cryotubes using a syringe, and immediately placed in liquid nitrogen. These were kept at −80 °C until analyses were performed. 

Blood collection was performed prior to feeding at 07:30 of days 27 and 28 of each period. Approximately 9 mL of blood were collected through brachycephalic venipuncture with 5 mL placed in a 5-SST yellow-cap tube for serum separation (BD Vacutainer™ tube BD 367989), and 4 mL was equally divided into two Hemogard tubes (grey cap) with EDTA (BD Vacutainer™ tube, BD 366450, BD Corporate, Franklin Lakes, NJ, USA). Tubes were centrifuged for 10 min at 3000 rpm (Allegra X-15R, Beckman-Coulter, Loveland, CO, USA) and kept on ice while processing. Plasma was separated for immediate complete blood count (XT-2000i, Sysmex, Landskrona, Sweden) and satiety panel analysis. A portion of the serum was used for blood chemistry (Cobas 6000, Roche Hitachi, Basel, Switzerland) and the remainder stored in cryotubes at −70 °C for LPS/LTA and metabolomics analyses.

#### 2.3.3. Food Intake and Fecal Analyses

For each animal, daily food intake was recorded and divided by the metabolic body weight (i.e., kg^0.75^) of each animal for normalization. These were averaged per animal per period before statistical analyses. Resistant starch intake was calculated based on RS determinations on each food [6]. Briefly, RS was the undigested starch fraction of a kibble after four hours of in vitro enzymatic incubation (K-DSTRS, Megazyme Inc., Wicklow, Ireland).

Fecal outcomes were determined on one stool defecation per dog per period. This fecal sampling scheme was considered representative of the period, as no abnormal stools were reported by trained technicians. Homogenized fresh feces from the feeding study were taken to the laboratory and fecal pH was measured immediately. Approximately 5 g of a thin layer of fresh feces were placed in an aluminum pan and dried at 104 °C for 3 h to determine moisture (modified AOAC 935.29). Feces were ground using a vibratory micro mill (Pulverisette 0, Fritsch Milling and Sizing, Inc., Pittsboro, NC, USA). The remaining dried feces were stored in liquid nitrogen until mineral analysis. Another portion of the fresh feces was placed in a porcelain crucible and heated in a muffle furnace for approximately 2 h at 600 °C to determine ash content (AOAC 942.05). For mineral analysis, dried feces were ground in the vibratory micro mill, acid digested (modified EPA 200.2) and minerals were determined in an Agilent 5100 OES system (Agilent Technologies, Santa Clara, CA, USA). Fecal ammonia was determined as described by [9].

#### 2.3.4. Satiety Hormones

Satiety hormones were measured in plasma samples collected before the provision of the day’s meal on d 27 and 28. The satiety panel included ghrelin, gastric inhibitory polypeptide (GIP), glucagon-like peptide 1 (GLP-1), glucagon, insulin, leptin, pancreatic polypeptide (PP) and peptide YY (PYY). Samples were evaluated for satiety hormones using an ELISA kit (Milliplex Map Canine Gut Hormone Magnetic Bead Panel—Endocrine Multiplex Assay; Millipore Sigma, Burlington, MA, USA), and measured on a Luminex 200 (Luminex Inc., Austin, TX, USA). 

#### 2.3.5. Microbiota: DNA Extraction, 16S Sequencing, PCR Amplification and Processing

Fecal samples freshly collected from the gut health study were frozen, cryo-extracted, then total DNA was extracted using a commercial kit (ZymoResearch quick-DNA fecal/soil microbe 96 magbead, Irvine, CA, USA). The PCR was performed by using forward (347F) and reverse (803R) primers bridging the V3-V4 hypervariable region of 16S RNA Amplicon, as well as Archeal forward (A349F) and reverse (A806R) primers (Life Technologies; Carlsbad, CA, USA). Amplicon sequencing was performed on a Illumina 16S metagenomic sequencing library, using a MiSeq system (Ilumina, San Diego, CA, USA). 

Output FASTQ files from the MiSeq instrument were pre-processed to contigs, chimeras were removed, and the highly variable V3V4 region of the 16S gene were filtered by the Mothur software [10]. Microbial taxa data sequences at the genus level were mapped with GreenGenes (V 13.5) [11]. Bacterial OTUs were normalized to copy number and further processed for functional profile using the Phylogenetic Investigation of Communities by Reconstruction of Unobserved States (PICRUSt) [12]. 

There were 231 total OTUs identified after high-throughput sequencing and copy number correction. The entire raw OTU count data were used to calculate alpha diversity indices as Hill numbers of order “*q*”, namely richness (*S*; *q* = 0), exponent Shannon (expH; *q* = 1), and inverse Simpson (invSimp; *q* = 2) [13,14]:$$D^{q}=\left(\overset{S}{\underset{i=1}{\sum }}Pi^{q}\right){\;}^{1/\left(1-q\right)}$$
where *S* is the number of microbial species, *Pi* is the relative abundance of the ith bacteria OTU, and parameter *q*, called the ‘order’ of the diversity measure, determines its sensitivity to species frequencies. Richness, *q* = 0; Exponent Shannon: *q* = 1, inverse Simpson: *q* = 2. 

Respectively, taxa richness (*S*) indicates the number of different taxa present in each fecal sample not considering their abundance; expH index has contribution of each taxa according to its abundance, and invSimp is mostly increased by the highly abundant taxa. Pielou’s evenness (J) was calculated as a fourth alpha diversity index [14]. Evenness J is a value constrained between 0 and 1 and provides insight on the homogeneity of abundances in a microbiota independent of the number of taxa. 

#### 2.3.6. Fecal Short-Chain Fatty Acids

Fecal fatty acids acetic, propionic, butyric, isobutyric, isovaleric and 2-methylbutyric were analyzed by a commercial laboratory (Metabolon, Morrisville, NC, USA). Briefly, fecal samples were spiked with a solution of eight stable labelled internal standards, homogenized, and subjected to protein precipitation. After centrifugation, an aliquot of the supernatant was derivatized. The reaction mixture was analyzed by LC MS/MS (Agilent 1290/AB Sciex 5500 system).

#### 2.3.7. Immunological Assays

Dog fecal immunoglobulin A (IgA) was quantified using a dog IgA ELISA quantitation set (Bethyl Laboratories, Montgomery, TX, USA). Gram-positive bacteria cell wall component lipoteichoic acid (LTA) was measured in dog serum by sandwich ELISA using a human lipoteichoic acid ELISA kit (Novateinbio, Woburn, MA, USA), and gram-negative bacterial endotoxin and membrane constituent lipopolysaccharide (LPS) was measured on dog serum by a turbidimetric assay using limulus amebocyte lysate (LAL) extract that reacts with the bacterial cell constituent (Pyrogent-5000 bulk kinetic turbidimetric LAL assay; Lonza, Mapleton, IL, USA). All immunological assays were performed at a commercial laboratory (MD Biosciences, Oakdale, MN, USA).

#### 2.3.8. Serum and Fecal Metabolomics

Analysis of plasma and fecal metabolites derived from metabolism of carbohydrates, protein, fat, vitamins, nucleotides, among others, were performed by a commercial laboratory (Metabolon). Metabolites were measured by four different methods using ultra-performance liquid chromatography (ACQUITY UPLC; Waters, Milford, MA, USA) and high resolution/accurate mass spectrometer interfaced with a heated electrospray ionization (HESI-II, Q-Exactive, Thermo Scientific, Waltham, MA, USA). Hydrophilic and hydrophobic compounds were analyzed using acidic positive ion conditions. The hydrophilic compounds were eluted using a water and methanol gradient-eluted C18 column (Waters UPLC BEH C18-2.1 × 100 mm, 1.7 µm) with 0.05% perfluoropentanoic acid (PFPA) and 0.1% formic acid (FA). The hydrophobic molecules were also eluted with a C18 column, but using methanol, acetonitrile, water, 0.05% PFPA and 0.01% FA. The third method consisted of basic negative ion optimized conditions with elution on a C18 gradient-eluted column using methanol and water with 6.5 mM ammonium bicarbonate at pH 8. Finally, the fourth method was negative ionization following elution from a hydrophilic interaction liquid chromatography (HILIC) column (Waters UPLC BEH Amide 2.1 × 150 mm, 1.7 µm) using a gradient consisting of water and acetonitrile with 10 mM ammonium formate at pH 10.8. Raw data ions were detected and processed using the laboratory library of standards for metabolite identification and for metabolite quantitation by peak area integration. Non-detected quantities were imputed with the observed minimum for detection for each individual compound. Furthermore, quality control of internal standards and endogenous metabolites ensured that results met the process specifications.

### 2.4. Statistical Analyses

#### 2.4.1. Preliminary Assessment of Dietary Digestibility

As part of the preliminary assessment of dietary treatments, the digestibility coefficients of HS and LS were performed on the same group of 6 dogs, and those of the MS were obtained from a separate group of 6 animals. Due to non-randomized design in this preliminary assessment, an unpaired *t*-test was conducted to compare HS vs. MS and MS vs. LS, and a paired *t*-test was conducted for the comparison between HS vs. LS treatments, using the TTEST procedure from Statistical Analysis Software (SAS) v 9.4 (SAS Institute, Cary, NC, USA). Diet palatability was analyzed as a paired *t*-test between intake ratio [HS intake ratio = HS intake/(HS + MS intake)] and intake ratio H1 = 0.5, using the TTEST procedure from SAS.

#### 2.4.2. Gut Health Study 

Response variables included food intake, fecal outcomes, blood chemistry and CBC, satiety hormones, fecal SCFA, gut integrity hormones, and alpha diversity indices generated from the microbiota data. For each response, a general linear mixed model was fitted to reflect the data generation process of a replicated Williams Latin Square design. Responses fecal ash, IgA, and butyric acid, as well as plasma GIP, ghrelin, insulin, PP, leptin, microbiota Pileou’s evenness and exponent Shannon, and intake (in both g and kcal per BW^0.75^) required a variance stabilizing transformation to meet model assumptions. In all cases, the linear predictor included the fixed effects of diet (LS, MS, HS), treatment sequence (1 to 6), period (1 to 3), and carryover effects specified with a sum-to-zero restriction. A random effect of housing building was fitted as an overarching blocking structure. A compound symmetry covariance structure was specified at the residual level for observations collected within a dog across periods. Response variables fecal acids (acetic, 2-methylbutyric, isobutyric and isovaleric), total BCFA, ash, ammonium, as well as serum LTA, glucagon and leptin, and microbiota inverse Simpson, richness and Pileou’s evenness, and RS intake required specification of heterogeneous residual variances. The best-fitting covariance structure for each analyte was selected according to Bayesian Information Criteria. Variance components were estimated using restricted maximum likelihood. Model assumptions were evaluated using externally studentized residuals and were considered to be reasonably met. A Kenward-Rogers approach was used to estimate degrees of freedom and make corrections to estimated standard errors. 

Statistical models were fitted using the GLIMMIX procedure of SAS Version 9.4. Estimated least square means, corresponding standard errors and 95% confidence interval are presented in the original scale. Pairwise treatment comparisons were conducted using a Tukey-Kramer approach to avoid inflation of Type I error rate due to multiple comparisons. Differences were considered significant at *p* < 0.05, and marginally significant at 0.05 < *p* < 0.10).

Fecal scores of dogs were analyzed as multinomial data assuming a cumulative logit distribution by the GLIMMIX procedure, with dog and period as random effects, and score frequency was computed by the FREQ procedure (SAS, v 9.4). 

#### 2.4.3. Metabolomics and Microbiota Data from the Gut Health Study

Metabolomics data and microbiota OTU count data were filtered and transformed prior to analyses. Specifically, metabolomics raw data was obtained as intensities of each analyte peak and were first median centered by dividing the datapoints by the median of each analyte, then converting to logarithm of base 2. Only analytes detected in over 80% of samples within at least one dietary group were included in the statistical analysis. The microbiota OTU data were specified at the family and genus levels. Similar to metabolomics, the OTU data was filtered so that all OTUs analyzed had counts above zero in more than 80% of fecal samples within at least one dietary group. Next, zero OTU counts were inflated using the Bayesian multiplicative treatment [15] and converted to centered log-ratio (CLR) to normalize the arbitrary counts determined by the DNA sequencer:(1)$$CLR\left(OTU_{X}\right)=Ln\left(\frac{OTU_{X}}{G_{OTU}}\right)$$
where OTU_X_ = counts of one OTU in the dataset; G_OTU_ = geometric mean of all OTU counts measured within a fecal sample. The CLR transformation converts each OTU count into a ratio relative to the geometric mean of all OTUs within each fecal sample [16]. Some properties of CLR are that the sum of all OTUs within a sample converge to zero (the mean), the reads are normalized to a common sequencing depth, and the CLR transformed values are scale invariant. This means that the same ratio is expected to be obtained in a sample with few read counts as a cohort sample with many read counts [16]. Before CLR transformation, OTU zero counts need to be inflated so that geometric means can be calculated. The Bayesian multiplicative approach prevents the most rare OTUs from being the most significant [16]. Finally, the robust outlier method correction Huber was applied to both OTU and Metabolomics datasets. The metabolomics serum and fecal data had 204 outliers removed altogether, and OTU data had 45. 

OTUs counts and serum and fecal metabolomics, each transformed as explained above, were analyzed using mixed models (JMP Pro v. 15). The linear predictor included the fixed effects for dietary treatment, period, treatment sequence and carryover A compound symmetry covariance structure was specified with period as the repeated structure and animal nested within sequence as the subject to allow for intraclass correlation of dogs across periods. Tukey-Kramer test was applied to correct for multiple comparisons. *p*-values for tests conducted on OTUs, serum and fecal metabolites were corrected for false discovery rate (FDR) [17] using the “qvalue” (R package; [18]). Treatment differences were considered significant when *q* < 0.10 and *p* < 0.05, and marginally significant when only *p* was less than 0.05. Sequential to statistical analysis, metabolites of both serum and feces were grouped according to the following classes: saccharolytic, proteolytic, energy metabolism, lipolysis, and bile salt metabolism for presentation and discussion purposes.

Finally, a Spearman Rho’s correlation was conducted between frequency of the significant microbiota OTUs (*p* < 0.05) (expressed in the centered log-ratio scale) and markers of carbohydrate fermentation, including SCFAs, fecal pH, fecal lactate and succinate, and fecal oligosaccharides. Response variables were binned into groups based on previously established criteria without regard to analysis. Each group was independently analyzed with a corrected *p* value with the corrected Spearman Rho *p*-values being adjusted according to the number of correlations within groups. The groups were OTUs vs. SCFA, OTUs vs. fecal sugars, and OTUs vs. fermentation markers; wherein, correlations were marked when the adjusted *p* value was significant (*p* < 0.05 for the group). This was defined by the individual *p*-value being *p* < (0.05/number of correlations within group).

## 3. Results

### 3.1. Digestibility and Palatability Assessment of Experimental Diets

Nutrient profile of the experimental diets met minimum and maximum levels for adult dogs at maintenance ([5]; Appendix A). There were not any problems reported with fecal quality in between sampling periods and on sampling days most scores were 5 (91.95%) and 4 (8.05%), though single fecal samples were missing for three dogs throughout the collection period (e.g., samples missing from one of the three diets for each of three dogs). There was no difference in dry matter (DM), organic matter (OM) and gross energy (GE) ATTD between treatments. (Table 1). 

Crude protein ATTD was approximately 2 percentage points greater for dogs fed LS compared to HS or MS, though there was no evidence for differences between the latter two. This dietary difference was not observed for digestibility coefficients of true protein ATTD. Also, crude fat ATTD was not significantly different among dietary treatments (*p* > 0.05). Finally, estimated ash ATTD was almost two-fold greater in LS diets compared to MS or HS diets (*p* < 0.05), though the latter two were not significantly different from one another. For daily intake and fecal outputs, on both as-is and DM basis, there was no evidence for any differences between dietary treatments. 

### 3.2. Gut Health Study

Dogs maintained body weight (BW) throughout the study and there was no evidence for a difference in mean BW among treatments (% body change of 0.033, 0.191 and −0.791, SEM = 0.4860, when dogs were fed the HS, MS and LS foods at different periods; *p* = 0.2297). No major adverse effects of diets were observed, and no dog was dismissed from the study. It is noted, though, that two animals received anti-inflammatory and antibiotic treatment during the third period due to injuries unrelated to the study. Overall, blood chemistry evaluations and CBC did not suggest any problems with overall health of dogs (Appendix A). As an exception, dogs fed the LS dietary treatment showed increased blood urea nitrogen (BUN) and BUN:creatinine (*p* < 0.05) relative to the other treatments, yet were within normal range (Appendix A). All fecal minerals were similar across treatments, except for sodium that was lower in feces of dogs fed the LS in comparison to HS (Appendix A).

#### 3.2.1. Food Intake and Fecal Outcomes 

Dogs in the LS dietary treatment had a lower caloric intake on a metabolic body weight (BW^0.75^) basis in comparison to the other treatments *(p* < 0.05; Table 2). Meanwhile, dogs fed MS and LS diets showed an increase in RS intake of approximately 45% and 60% RS (*p* < 0.01) relative to dogs fed HS, respectively (Table 2). 

Most fecal samples (65 out of 71) were firm and considered normal (scores 4 and 5), and only a small proportion had a score 3 (softer); no evidence for treatment differences was detected (*p* = 0.23; Figure 1). Throughout the study, stool samples had no evidence for treatment differences (*p* = 0.22 and 0.25, respectively). Fecal pH had marginal evidence for a decrease in dogs fed LS dietary treatment compared to HS (*p* = 0.07); this may be indicative of more carbohydrate fermentation. There was no evidence for any treatment effect on fecal ammonium (*p* = 0.20) nor on fecal ash excretion (*p* = 0.10). 

#### 3.2.2. Microbial Fermentation Products, Endotoxemia and Immunity

For total fecal SCFA, acetic, or propionic acids concentrations, there was no evidence for any effect of dietary treatment (*p* > 0.10; Table 3). By contrast, dietary effects were apparent on fecal butyric acid concentration (*p* = 0.02); whereby, dogs fed the LS diet had approximately 37% greater fecal butyric acid than those fed the HS diet. Meanwhile, fecal butyric acid concentrations under the MS dietary treatment were intermediate to the other two diets and not significantly different from either. The approximate proportions of acetic:propionic:butyric acid in the HS, MS and LS dietary treatments were calculated as 47:32:22, 44:29:28 and 43:29:28, respectively. 

Branched-chain fatty acids may be attributed in part to protein putrefaction. In general, there was no evidence for any dietary effect on fecal production of BCFA (*p* > 0.35; Table 3). 

Gut wall integrity was assessed by fecal immunoglobulin A and serum concentrations of gram-positive and gram-negative bacteria cell wall components (LTA and LPS, respectively). There was no evidence of any dietary effect on fecal IgA or serum LTA (*p* > 0.61; Table 3), but LPS was marginally increased (*p* = 0.08) under the MS and LS dietary treatments compared to HS.

#### 3.2.3. Satiety Hormones

When satiety hormones were measured in dog plasma, there was one sample which had concentrations below the limit of detection for every hormone and was therefore excluded from analyses. Statistical analyses of satiety hormones were performed on data from 23 plasma samples. For concentrations of ghrelin, leptin, GIP, glucagon, PP and PYY, there was no evidence for any dietary effect (*p* > 0.05; Table 4). However, there was marginal evidence for a treatment effect on insulin (*p* = 0.06) concentration, whereby dogs fed the LS dietary treatment tended to have greater levels of insulin at fasted state than those fed the HS. Glucagon-like peptide 1 had 35 samples (out of 72) below the limit of detection, so this hormone was not analyzed, and results are not reported here.

#### 3.2.4. Metabolomics

A total of 832 fecal metabolites and 858 serum metabolites were detected in the samples, and 80% fecal and 88% serum analytes passed the 80% filter described under Materials & Methods. The majority of treatment differences were identified in fecal metabolites rather than in serum metabolites. For fecal glucose, maltose and maltotetraose, significant dietary effects were apparent (*p* < 0.05; *q* < 0.10; Figure 2.); while dogs fed the LS and MS foods had greater levels of fecal maltose and maltotetraose than those fed the HS diet, and fecal glucose was greater in the LS than the HS group, with MS similar to the extremes. There was also more succinate in feces of dogs fed the LS relative to HS food (*p* < 0.05; *q* < 0.10; Figure 2), which may indicate greater saccharolytic activity in dogs fed the LS food. 

For fecal glutamate, glycine and threonine, and dipeptides phenylalanylalanine and tryptophylglycine, there was marginal evidence for greater levels in dogs fed the less processed foods (MS and LS) relative to HS (*p* < 0.05; *q* > 0.10; Appendix A). Seven putrefactive fecal compounds were affected by dietary treatment (*p* < 0.05; *q* < 0.10; Appendix A). Specifically, 3-hydroxyindolin-2-one and indole decreased in the LS and MS treatments relative to HS; phenol sulfate and indolin-2-one decreased only in the MS treatment relative to HS; 3-indoleglyoxylic acid increased only in dogs fed the LS food; and finally, indolepropionate and 2-hydroxy-3-methylvalerate increased in dogs fed the MS relative to HS (*p* < 0.05; *q* < 0.10; Appendix A). We also found evidence to support an increase in fecal pyrraline (*p* < 0.05; *q* < 0.10; Appendix A) in dogs fed the MS and LS foods in comparison to HS. Pyrraline belongs to the group of advanced glycation end-products (AGEs). This finding was unexpected as these foods were processed in a similar manner except for extruder shaft speed and in-barrel moisture. The development of AGEs are associated with over-processed foods [19]. 

Evidence for dietary treatment effects on fecal metabolites was particularly apparent in components of lipid metabolism. For instance, fecal long chain mono-unsaturated and poly-unsaturated fatty acids (LCMUFA and LCPUFA) were overall higher in dogs fed the HS food (Figure 3A) compared to LS and MS. Specifically, two LCPUFAs, stearidonate (18:4n3) and eicosapentaenoate (EPA; 20:5n3), were higher in the HS relative to MS group (*p* < 0.05; *q* < 0.10), and arachidonate (20:4n6) showed marginal evidence (*p* < 0.05; *q* > 0.10) for an increase in dogs fed the HS relative to the LS diet (Figure 3A). Conversely, molecules from intermediate fatty acid metabolism, namely palmitoyl-linoleoyl-glycerol (16:0/18:2), diacylglycerol (16:1/18:2, 16:0/18:3), linoleoyl-linoleoyl-glycerol (18:2/18:2) and linoleoyl-linoleoyl-glycerol (18:2/18:2), were increased overall for dogs fed the MS dietary treatment (Figure 3B). More specifically, diacylglycerols palmitoyl-linoleoyl-glycerol (16:0/18:2) and linoleoyl-linoleoyl-glycerol (18:2/18:2) were increased under the MS dietary treatment relative to HS and LS, and isomer linoleoyl-linoleoyl-glycerol (18:2/18:2) and diacylglycerol (16:1/18:2, 16:0/18:3) were greater in the MS dogs fed dietary treatment in comparison to LS, with no evidence for any differences compared to HS (*p* < 0.05; *q* < 0.10). There was also marginal evidence (*p* < 0.05; *q* > 0.10) favoring oleoyl-linoleoyl-glycerol (18:1/18:2) and palmitoyl-linoleoyl-glycerol (16:0/18:2) two diacylglycerols in dogs fed the MS food.

The primary bile salt taurocholate measured in dog feces was more abundant (*p* < 0.05; *q* < 0.10) when dogs were fed the MS or LS diets compared to HS (Figure 3C). Meanwhile, three products of secondary bile salt metabolism dehydrolithocholate, lithocholate and taurodeoxycholate were greater in feces of dogs fed the HS diet relative to MS or LS (*p* < 0.05; *q* < 0.10). Taken together, these findings seem to suggest that dogs fed the more processed diet (i.e., HS) had more bacterial metabolism of bile salts compared to MS and LS. 

Many of the serum long-chain saturated (LCSFA), mono- (LCMUFA) and poly-unsaturated (LCPUFA) fatty acids had an overall pattern of lower concentrations in dogs fed the MS diet compared to either of the LS or HS alternatives (*p* < 0.05; *q* > 0.10; Figure 4). This was expected, as dogs fed the MS diet also had a trend for lower concentrations of LCMUFA and LCPUFA in feces. 

#### 3.2.5. Microbiota

##### Alpha Diversity

There was no evidence for any dietary effect on alpha diversity taxa richness, Pileou’s J evenness, inverse Simpson or exponent Shannon (*p* > 0.74 in all cases; Table 5). This would indicate that the number of bacterial OTUs identified, the relative evenness of their abundances and the complexity of the community with regards all taxa as well as highly abundant taxa (respectively) were not significantly different between dietary treatments.

##### OTU Centered Log-Ratio (CLR) Transformed

Ninety OTUs passed the 80% filter before conducting statistical analysis of OTUs with a centered log-ratio (CLR) transformation technique. Only OTUs that presented *p* < 0.06 for the overall test on treatment effects are reported on Table 6, which lists estimated mean OTU counts expressed in the CLR scale at the family and genus levels, sorted primarily by Phylum and secondarily by their decreasing order of abundance. When genus was unclassified, OTU assignment was considered ambiguous and thus inconclusive, as it could be considered of any genus of that given family. Meanwhile, a missing genus indicated that there was no further annotation available.

Some of the most abundant OTUs had no evidence of any dietary effects, including Prevotellaceae Prevotella, Clostridiaceae Clostridium, Lactobacillaceae Lactobacillus, Alcaligenaceae Sutterella, Lachnospiraceae Ruminococcus and Fusobacteriaceae Fusobacterium. In contrast, the most abundant OTUs for which significant treatment effects were identified, were within Phylum Firmicutes. Specifically, Lachnospiraceae Blautia was the most abundant OTU with a marginal significance (*p* = 0.06; *q* > 0.10); wherein, dogs fed the HS diet had higher abundance of this microbe than those fed the MS food, while dogs fed LS were not different from MS or HS (Table 6). This OTU also had a negative correlation with total SCFA and butyric acid (Table 7). As for Turicibacteraceae Turicibacter and Veillonellaceae (no further annotation for genera), both were found to be less abundant in dogs fed the MS diet than the HS diet, whereas dogs fed LS were not different from MS and HS. Meanwhile, Lachnospiraceae Roseburia and Erysipelotrichaceae Catenibacterium were more abundant in MS-fed dogs than those fed HS, again with LS-fed dogs showing intermediate abundance estimates (Table 6). Catenibacterium had a positive correlation with fecal glucose (Table 7).

Third in abundance was Coriobacteriaceae Slackia, which was the only representative of phylum Actinobacteria and showed significant treatment effects on OTU abundance (Table 6) [within marginally significant OTUs (*p* < 0.05, *q* > 0.10)]. Much like Blautia and Turicibacter, Slackia abundance was also greater in HS-fed dogs compared to MS-fed, with LS showing no significant differences from either (*p* < 0.05; *q* > 0.10), and negatively correlated with total SCFA (Table 7). 

Bacteroidaceae Bacteroides abundance was above the average (0.0 on CLR scale) and were marginally significant (*p* < 0.05; *q* > 0.1). Dogs fed the MS diet had greater abundance of this OTU than HS-fed dogs, with LS showing intermediate estimates (Table 6). Bacteroides had a strong and positive correlation with butyric acid (Table 7). Based on evidence from this study, the genus Bacteroides is considered saccharolytic. Porphyromonadaceae with unclassified genera (ambiguous classification) was the only OTU in the dataset with evidence of a dietary effect (*p* < 0.05; *q* < 0.10) and its abundance was above the average (0.0 on CLR scale). This OTU was more abundant in dogs fed the HS than those fed the MS and LS diets. The low-abundance bacteria that belonged to phylum Bacteroidetes (*p* < 0.05; *q* > 0.1; Table 6), Paraprevotellaceae (without genera annotation) and Prevotellaceae unclassified (with ambiguous genera annotation) had lower abundance in the HS and LS treatments relative to MS, and both had a strong correlation with butyric acid (Table 7). 

Overall, OTU abundance in feces of dogs fed the MS food had greater separation from those fed the HS diet, and LS was intermediate. This was contrary to our expectation that the LS diet would lead to a higher separation of OTU abundance relative to the HS diet, but this was only observable for one OTU, namely Enterobacteriaceae Yersinia. 

## 4. Discussion 

The term resistant starch (RS) refers to the starch fraction that escapes digestion by mammalian enzymes in the small intestine and reaches the large intestine where it is fermented by commensal bacteria into beneficial energy substrates like SCFAs [20]. The resistance to digestion of these starches can be by means of cell walls, of their raw granular crystalline format, retrogradation, chemical modification, or complexation with lipids (RS types I, II, III, IV and V, respectively; [21]). In previous work on this topic, dog kibbles with three levels of thermomechanical energy were produced through a single-screw extruder (Wenger X115, Sabetha, KS, USA) by modifying extruder in-barrel moisture (IBM) and shaft speed (SS; [6]) with the goal to retain more RS types II and III. These diets were fed to dogs in the present study. Our hypothesis was that canine consumption of foods having increased retention of resistant starch via modulation of thermomechanical energy would lead to increased levels of gut bacteria with documented metabolic predilections toward saccharolysis. Secondarily, we sought to assess the meaningfulness of microbial changes by assessing whether fecal products of saccharolytic fermentation (e.g., SCFA, pH, lactate) changed in a manner concordant with the fecal microbiota community abundances. We found that although there were relatively minor changes to the fecal microbiota abundances and community structure, there were more numerous and larger magnitude changes in the products of gut microbiota fermentation of carbohydrate. These data show the metabolic flexibility of the canine gut microbiota and emphasize that altering substrate availability may mediate positive changes in levels of saccharolytic postbiotics without necessarily changing which bacteria are present. Thus, data from the present work point towards a change in microbial activity without significantly changing the microbial ecology.

The amount of starch that escapes small intestinal (SI) digestion and reaches the colon can vary according to dietary effects such as food matrix composition, starch processing and cooking level, and individual dog factors such as food intake, species mastication habits, transit time, among others [20]. Resistant starches that are not chemically modified can be digested in the small intestine, given the right amount of enzymes relative to substrate and adequate environmental conditions [22]. 

There might be a concern that high levels of RS would cause digestive upset due to overfermentation. However, this was not the case of the present study because the amount of RS in each diet was low (<1.06%; Appendix A). Previous reports indicate that small-sized dogs had no adverse stool quality issues even when fed as much as 7% RS per day [23]. Further, there was no expectation for diet refusals since corn is known to be palatable for the dog [24].

Processing inputs to produce dietary treatments in this experiment were intentionally modified to produce increasing levels of slowly digestible starch (SDS) and RS as determined by an in vitro enzymatic procedure [6]. While the intent was to alter RS, the more appropriate description of dietary treatments was high, medium and low shear, because what we could control was the mechanical energy input in each diet. The high shear food was produced similar to most pet foods available in the market with the highest level of starch gelatinization and lowest amount of RS or SDS [6]. By contrast, the low shear food was produced with less mechanical energy and retained more RS and SDS, because starch was cooked to a lesser extent than the other diets [6]. Meanwhile, the MS diet was intended to be intermediate, though results indicated that it behaved more like the LS food with regard to changes in the microbiota, SCFAs and fecal sugars. The net result of increasing RS would be a bypassing of starch from the SI into the large intestine/colon for fermentation. Thus, this RS might be considered a prebiotic, defined as “selectively fermented ingredients that allow specific changes, both in the composition and/or activity in the GI microflora that confer benefits upon host wellbeing and health” [25]. In our study, the increased fecal butyric acid when dogs were fed the LS relative to HS implies a potential improvement in the gut health which is typical of that of a prebiotic.

When fed to dogs, the two diets produced with less thermomechanical energy, namely MS and LS, resulted in many physiological and biochemical profiles that did not differ significantly from one another. Ash digestibility on the preliminary assessment was greater when dogs were fed the diet containing more RS (LS food), which corroborates findings from other studies [26,27] regarding an increased absorption of some minerals when prebiotics were added to the dog and human foods. Both the LS and MS diets induced increased fecal glucose and oligosaccharides than the HS diet, which was consistent with the work of [2] for dogs fed a low shear food based on corn and rice. This finding provided indirect confirmation that the treatments produced with lower thermomechanical energy, namely LS and MS, were likely effective in retaining RS since glucose and glucose-based oligosaccharides are starch derivatives. The small intestine (SI) is sensitive to dietary changes and has a higher abundance of carbohydrate fermenting bacteria than the colon [28,29]. Thus, considering that feces of dogs fed the LS or MS diets had more oligosaccharides than those fed the HS it would suggest the presence of higher concentrations of these sugars in both the SI and proximal regions of the large intestine.

The gastro-intestinal tract (GI) harbors trillions of metabolically active bacterial species that compose the microbiota, as well as a small percentage of fungi, archaea, protozoa and virus [29,30]. The gut microbiota in mammals are known to be commensal, which in Latin means “sharing a dining table” [31]. This definition was attributed to these bacteria because they live in symbiosis with their host. Specifically, gut commensal bacteria can utilize host endogenous molecules or dietary products that bypass upper digestive tract digestion and generate fermentation by-products or post-biotics that benefit host health [32]. By contrast, in the event that there is a stressor and the luminal environment is perturbed with overgrowth of pathological microbes, there may be dysbiosis that can negatively affect the body system [32]. Rather, the increase in starch fermentation byproducts, specifically butyric acid, suggested that both the LS and MS diets likely improved gut health. That the clinical blood work and stool quality stayed within normal ranges supports the conclusion that RS did not have a significant negative impact on dog health.

The most predominant phylum in the gut of healthy dogs are reportedly Bacteroidetes, Firmicutes, Proteobacteria, Fusobacteria, and Actinobacteria [29,33,34], which is consistent with results from this study. Firmicutes were the most abundant phylum, with some bacteria exhibiting saccharolytic activity and others not. The growth of species belonging to the phylum Firmicutes were favored by a mildly acidic pH in the colon, and dogs fed more RS (i.e., LS and MS) tended to have lower fecal pH (due to fermentation). Ref. [35] reported that there was a high proportion of Firmicutes (51%) attached to resistant starch in the human colonic microbiota. Anaerobic gram-positive Roseburia within the phylum Firmicutes has the ability to utilize starch and produce butyric acid [20,36]. However, findings from this study indicate that this microbe was not correlated to butyric acid or SCFA. Genus Blautia was the most prevalent OTU in our dataset and correlated negatively to both total SCFA and butyric acid. Studies in other species have reported Blautia to have a negative correlation to carbohydrates [37,38]; whereas, [39] observed an increase in Blautia in the cecum and colon of pigs fed a retrograded potato starch (RS source) relative to corn starch. 

Bifidobacterium from phylum Actinobacteria has been the focus of human studies due to their effectiveness in utilizing starches as energy substrate [40]. However, the only marginally significant representative of phylum Actinobacteria in our dataset was Slackia, and it had a negative correlation to SCFA. Although our data did not indicate any evidence for saccharolytic activity based on Spearman rho correlations, Slackia has been previously reported to be saccharolytic and to increase with fermentable fiber addition to the diet [26]. The Bacteroides genus was found to have broad saccharolytic potential which agrees with the other research reports in the literature [20,26,41]. The present study provided evidence that Bacteroides increased in RS-rich diets, that is MS and LS. Most members of phylum Bacteroidetes had a high saccharolytic activity with butyric acid producing capacity. Bartonella could possibly be pathogenic and had a tendency to decrease in dogs fed the LS and MS foods. Ref. [26] reported that a high meat food supplemented with a fiber blend led to a decline in Bartonella. A second study from the same group reported an increase in Bartonella in dogs fed a low shear diet, that is a food with elevated RS [2]. The latter study also reported an increase in Yersinia in dogs fed the low shear diet [2], which is consistent with our results. 

Important tools to analyze microbial compositional changes are alpha diversity indices, which are mathematical measures of species diversity in a community that provide information about rarity or commonness of different species [14]. Alpha diversity is characterized as the variation in the microbiota within each dog. There are no clear definitions between an ecosystem diversity and its health as their validity are limited within each system [42]. For example, when studying forests one can find environments with low diversity measures, but that are productive, healthy and have integrity, whereas others can be highly diverse but with low stability and productivity [42]. This analogy can be translated to the dog microbiota. It is difficult to characterize a specific microbiota profile for health or disease, because there is large variation between individuals’ microbiotas [29,43,44]. The functional profile of the microbiota is usually a better assessment of health as compared to the microbiota composition [29]. One can determine the health of a microbiota by measuring biomarkers known to positively affect the animal, such as SCFAs. In the present study, there was no evidence for differences in alpha diversity though the relative abundance of specific OTU seemed to differ between dietary treatments. Consistent with our work, [45] also observed that the addition of prebiotics to a dog food failed to induce shifts in alpha diversity in the gut, as assessed in fecal samples. However, some fermentation products including SCFAs increased in response to prebiotics addition to the food and proved these to be beneficial to the colonic health of dogs. In our study, in order to detect differences in alpha diversity it might have been necessary to extend the adaptation time (over 4 weeks), or to produce foods with lower mechanical energy during the extrusion process to enhance RS concentration. Indeed, when dogs were fed a low shear food after 6 weeks of feeding, an increase in species richness was observed compared to dogs fed a high shear food [2]. In their work, the difference in mechanical energy between the low and high shear foods was 4-fold, while in our study, this difference was only 1.7-fold. 

Large polysaccharides that reach the colon are first hydrolyzed by primary polysaccharidases which release smaller oligosaccharides that serve as substrate for fermentation and ultimately production of SCFAs [46]. A small proportion of SCFAs are also derived from protein fermentation [47]. Bacterial enzymes that degrade starch comprise glycoside hydrolases [48]. Starch α-1,4 linkages can be catalyzed by bacterial α-amylase or α-glucosidases, while branching-point α-1,6 linkages are hydrolyzed by a pullulanase [20]. Bacteria also contain binding domains from different families that are responsible for their adhesion to starch molecules as the first step to the degradation process [49]. 

Once the microbiota hydrolyzes fermentable fiber or resistant starches into monosaccharides, these are fermented by bacterial enzymes in the anaerobic environment of the colon. The major bacterial metabolic route for six-carbon sugars such as glucose is the Embden-Meyerhof-Parnas pathway [50]. Ref. [51] provided a good review on pathways to produce SCFA to further understand the metabolism involved in their synthesis. In short, glucose is first fermented to pyruvate, which is reduced to lactate and ethanol. The production of SCFA is linked to the attempt of bacteria to decrease reducing agents (such as 2H+ and NADH) in their environment. A major part of pyruvate can be converted to acetyl-CoA with the formation of H_2_ and CO_2_. Acetic acid can be formed from hydrolysis of acetyl CoA or from CO_2_, while propionic acid can be produced from the reduction of lactate. There is another metabolic pathway to produce propionic acid from the pentose-phosphate pathway, which is most relevant in fermentation of fibers rather than RS. Butyric acid synthesis starts by linking two acetyl-CoA molecules, which can derive from pyruvate or acetic acid [51]. Butyric acid can also be produced by lactate-utilizing bacteria [52]. The LS and MS foods in the present study favored higher production of fecal butyric acid. Butyric acid has been a preferred end-product for studies of prebiotics [53]. 

Butyric acid is mainly used as colonocyte energetic substrate, but also presents anti-inflammatory properties that improve intestinal homeostasis and mucosa immunity [30,54]. All SCFAs; butyric, propionic and acetic acids, have been reported to regulate satiety in the long-term via activation of PYY and GLP-1 and increased expression of leptin, exerting an anorexigenic effect [54]. These satiety effects may be attributed mainly to propionic and acetic acids, since butyric acid is mostly utilized at the intestinal level [55]. In the present study there was no evidence for a dietary treatment effect on propionic and acetic acids concentrations, nor on satiety in the long term. It is possible that there wasn’t enough RS to promote satiety through SCFA production, and (or) that dogs were fasting for a longer time (>20 h) which overrode changes in satiety hormones. In the present study, the MS and LS diets did not show any evidence for improvement on markers of colonic immunity in fasted dogs relative to the HS treatment. Likewise, [3] did not find increases in fecal IgA when dogs were fed a high RS (type II) diet from corn. However, [2] reported greater amounts of epithelial sugar and fecal IgA after feeding a low shear food to dogs which indicated a faster epithelial cell turn over and improved local immunity. These findings may be partially explained by a longer feeding period and a lower mechanical energy to produce the food than in the present study (specific mechanical energy 9 vs. 23.6 Wh/kg, respectively; [6]).

The main reduced fermentation byproducts include lactate, succinate, H_2_, and butyric acid [40]. Dogs fed the LS or MS diets showed marginal evidence for an increase in lactate, succinate and butyric acid, and decreased fecal pH, which were indicative of starch fermentation by saccharolytic bacteria [51,56]. Ref. [3] also reported an increase in fecal lactate, butyric acid, and a decrease in pH associated with low shear (high RS) food consumption. Luminal pH declines from the ileum to the colon due to higher SCFA production [51]. The drop in pH in the large intestine is important for shifting the microbiota to prevent overgrowth of pathogenic strains [51,57]. The SCFAs are absorbed by colonocytes in exchange for bicarbonate (HCO_3_^−^), which acts as a buffer and increases the pH again as digesta passes to the rectum [51]. So, it is possible that dog colonic pH would be lower than what was measured in the rectum or feces. A similar phenomenon occurs with SCFAs. The concentration of SCFAs was much higher in the proximal regions of the swine colon compared to distal portions [58], which would be closer in composition to the feces excreted. Only ~5% of total SCFA produced are present in the feces with most butyric acid absorbed by the epithelial cells [47,59]. Although the butyric acid difference in the present study seemed small, the actual production could be much greater. Moreover, when carbohydrate substrates become less available at the distal colon, there is a downward shift in butyric acid producing bacteria and propionic and acetic acids producing strains become more prevalent [60]. Other studies have also reported that dogs and cats fed an extruded food with RS from corn had increased fecal butyric acid [2,3,4,61].

Butyric acid has an important role in suppressing colonic carcinogenesis and regulating gut immunological homeostasis [31]. Ref. [58] found that RS was completely degraded in the swine cecum and that butyric acid was increased in the proximal portion of their colon. The digestive system of pigs is larger than canines and possesses a greater fermentation capacity. Butyric acid administered orally also prevented mice fed a high fat diet to develop insulin resistance and obesity [62].

While there were significant changes in carbohydrate metabolism by the gut microbiota of dogs fed the LS and MS foods, analogous outcomes were not observed in serum. Metabolomics detection of analytes are directly tied to the time of blood collection. Since dogs were fasting and blood glucose levels are constantly being controlled by insulin, glucagon and other hormones, we did not expect changes in either carbohydrate or Krebs cycle metabolites in serum, and this was confirmed.

A possible route for energetic metabolism that sustains long term energy supply is through fatty acid oxidation which is activated by SCFAs concurrent with inhibition of lipid synthesis from glucose [63]. In the present study there were some trends in metabolomics of lipids. The increase in diacylglycerols in feces of dogs fed the MS food could be explained by the higher bypass of these lipids to the colon. Interestingly, serum of these dogs had lower levels of some fatty acids than the other treatments. This suggests that bacterial breakdown of diacylglycerols into fatty acids and consequent absorption into the blood stream were less pronounced in the MS treatment. Ref. [64] reported that mice fed corn diets supplemented with chemically modified RS had lower body weight gain, as well as lower levels of serum total lipids, triglycerides, and cholesterol. Some studies have reported reduced serum triglyceride associated with RS type II consumption by healthy humans [65]. In the present study, dogs fed the MS food had lower levels of serum fatty acids, but there was no evidence for differences amongst treatments in serum triglycerides or cholesterol measured by blood chemistry. Conversely, dogs fed the HS diet had a tendency for more fecal long-chain polyunsaturated fatty acids (LCPUFA) than the other treatments, though the LCPUFA serum concentrations were not significantly different from the LS treatment. This would be expected since long-chain fatty acids are not readily absorbed across the colon in a quantitatively meaningful manner, so serum LCPUFAs should not be directly correlated with its amount in feces. Saccharolytic bacteria such as Bifidobacteria, Roseburia and Lactobacillus are responsible for the breakdown of LCPUFAs into conjugated linoleic acid (CLA) [63] which could partially explain the decreased fecal LCPUFAs observed when dogs were fed the MS and LS diets in our study. The tendency to decrease fecal medium chain fatty acids (MCFAs), as well as some mono- and diacylglycerols when dogs were fed the HS food may suggest that the restriction of fermentable carbohydrate induced microbes to seek alternative carbon sources.

The high shear food favored the increase of both primary and secondary bile salts in dog feces. Bile salts need to be deconjugated by microbial bile salt hydrolase (BSH) in order to not be reabsorbed and bypass to the colon [66]. The activity of this enzyme is greater in gut microbes residing in the terminal ileaum and colon. There is large variation of BSH between bacterial species and these can be transmitted between microbes through plasmids [66]. The increase in bile salt excretion contributes to a decrease in serum cholesterol as it needs to be used to replenish the lost liver bile salts. In parallel, the fraction of secondary bile salts that are reabsorbed by the host modulates the systemic lipid and glucose metabolism, contributing to an improved liver and pancreatic functions, as well as improved glucose tolerance [63]. Taurocholate was greater in feces of dogs fed the high shear (less RS) food of the present study. This agrees with observations in dogs reported by [2]. Some secondary bile salts also increased in feces of dogs fed the HS food, which may suggest a higher microbial metabolism of bile salts in this treatment. Conversely, [2] reported that four secondary bile salts increased in dogs that consumed the low shear foods, and six that increased for dogs fed a high shear treatment. The mechanism that led consumption of the high shear food to increase bile salt excretion is not easily explained.

## 5. Conclusions

To our knowledge this is the first work to explore in depth the systemic effects of foods extruded at three levels of mechanical energy in the dog down to the level of the microbiota and metabolomics. Most positive changes resulted from dogs fed the LS and MS treatments relative to the HS food. Although there was no evidence for any treatment differences in bacterial alpha diversity in fecal microbial populations of fed dogs, there was evidence to support growth of some saccharolytic bacteria favored by consumption of the LS and MS foods. This also resulted in more fecal glucose and oligosaccharides than the HS treatment. By-products of carbohydrate fermentation succinate, butyric acid, lactate and H_2_ (measured as fecal pH) confirmed that carbohydrate metabolism was more extensive in dogs fed diets produced at low and medium shear. The most relevant health biomarker that increased with the consumption of the LS and MS foods was fecal butyric acid. It has been identified in previous research to be the major player in providing energy substrate for colonocytes, preventing colonic cancer, and decreasing local and systemic inflammation in monogastric animals. Based on evidence from this work, we conclude that a low to medium shear food (elevated RS) might benefit gastrointestinal and systemic health of dogs.

## Figures and Tables

**Figure 1 microorganisms-09-02293-f001:**
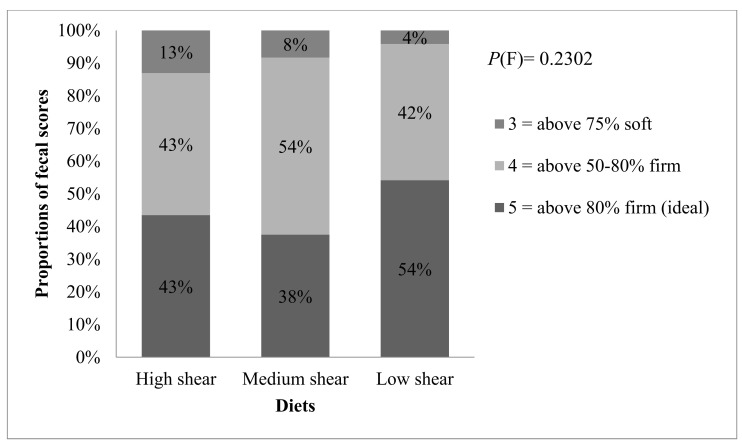
Relative proportions of stool scores for dogs fed dietary treatments produced at high, medium and low shear diets (*n* = 23, 24 and 24, respectively).

**Figure 2 microorganisms-09-02293-f002:**
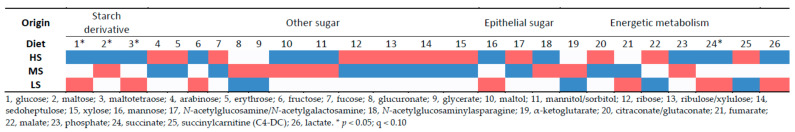
Heatmap of fecal saccharides metabolomics of dogs (*n* = 24) fed diets produced at high, medium and low shear (HS, MS and LS, respectively). Specific analytes were numbered 1 to 26 and grouped as starch derivatives, other sugars, epithelial sugars and energetic metabolism. Estimate treatment means expressed in the Log_2_ scale were color coded within each metabolite across diets on a color scale. Blue indicates a relative concentration lower than the other treatments, white indicates an intermediate relative concentration, and red indicates a higher relative concentration than the other treatments.

**Figure 3 microorganisms-09-02293-f003:**
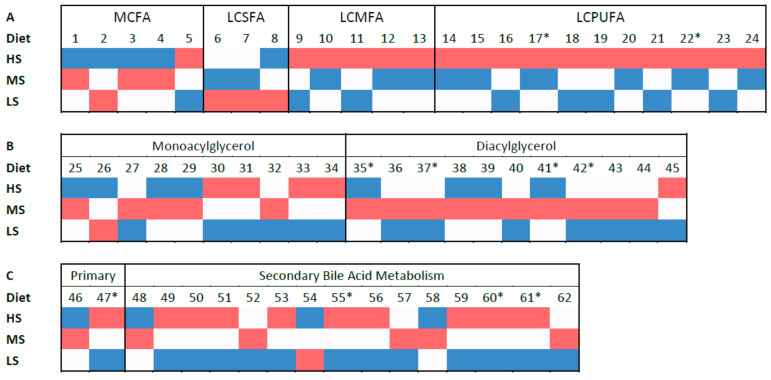
Fecal lipid metabolism analytes heatmap of dogs (*n* = 24) fed diets produced at high, medium and low shear (HS, MS and LS, respectively), with each individual group of analytes ((**A**): MCFA, medium-chain fatty acids; LCSFA, long-chain saturated fatty acids; LCMUFA, long-chain mono-unsaturated fatty acids; LCPUFA, long-chain poly-unsaturated fatty acids. (B): monoacylglycerols and diacylglycerols. (**C**): primary and secondary components of bile salt metabolism). Log_2_ values were conditionally formatted within each metabolite across diets on a color scale. Blue indicates a relative concentration lower than the other treatments, white indicates an intermediate relative concentration, and red indicates a higher relative concentration than the other treatments. (**A**): 1, caproate (6:0); 2, heptanoate (7:0); 3, caprylate (8:0); 4, caprate (10:0); 5, 5-dodecenoate (12:1n7); 6, palmitate (16:0); 7, stearate (18:0); 8, arachidate (20:0); 9, palmitoleate (16:1n7); 10, 10-heptadecenoate (17:1n7); 11, oleate/vaccenate (18:1); 12, 10-nonadecenoate (19:1n9); 13, erucate (22:1n9); 14, hexadecadienoate (16:2n6); 15, hexadecatrienoate (16:3n3); 16, linoleate (18:2n6); 17, stearidonate (18:4n3); 18, docosadienoate (22:2n6); 19, mead acid (20:3n9); 20, adrenate (22:4n6); 21, arachidonate (20:4n6); 22, eicosapentaenoate (EPA; 20:5n3); 23, docosapentaenoate (n6 DPA; 22:5n6); 24, docosahexaenoate (DHA; 22:6n3). (**B**): 25, 1-palmitoylglycerol (16:0); 26, 2-palmitoylglycerol (16:0); 27, 1-palmitoleoylglycerol (16:1); 28, 2-palmitoleoylglycerol (16:1); 29, 1-heptadecenoylglycerol (17:1); 30, 1-oleoylglycerol (18:1); 31, 2-oleoylglycerol (18:1); 32, 1-linoleoylglycerol (18:2); 33, 2-linoleoylglycerol (18:2); 34, 1-linolenoylglycerol (18:3); 35, palmitoyl-linoleoyl-glycerol (16:0/18:2); 36, palmitoyl-linoleoyl-glycerol (16:0/18:2); 37, diacylglycerol (16:1/18:2 [2], 16:0/18:3 [1]); 38, oleoyl-oleoyl-glycerol (18:1/18:1); 39, oleoyl-linoleoyl-glycerol (18:1/18:2); 40, oleoyl-linoleoyl-glycerol (18:1/18:2); 41, linoleoyl-linoleoyl-glycerol (18:2/18:2); 42, linoleoyl-linoleoyl-glycerol (18:2/18:2); 43, linoleoyl-linolenoyl-glycerol (18:2/18:3); 44, linoleoyl-linolenoyl-glycerol (18:2/18:3); 45, linolenoyl-linolenoyl-glycerol (18:3/18:3). (**C**): 46, cholate; 47, taurocholate; 48, 12-dehydrocholate; 49, 12-ketolithocholate; 50, 3b-hydroxy-5-cholenoic acid; 51, 3-dehydrodeoxycholate; 52, 6-oxolithocholate; 53, 7alpha-hydroxycholestenone; 54, 7-ketodeoxycholate; 55, dehydrolithocholate; 56, deoxycholate; 57, deoxycholic acid 12-sulfate; 58, hyocholate; 59, isohyodeoxycholate; 60, lithocholate; 61, taurodeoxycholate; 62, ursocholate. * *p* < 0.05; *q* < 0.10.

**Figure 4 microorganisms-09-02293-f004:**
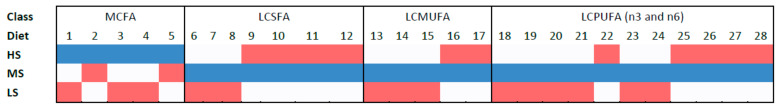
Serum non-esterified fatty acids heatmap of dogs (*n* = 24) fed diets produced at high, medium and low shear (HS, MS and LS, respectively), with each individual group of fatty acids (MCFA: medium-chain fatty acids; LCSFA: long-chain saturated fatty acids; LCMUFA, long-chain mono-unsaturated fatty acids; LCPUFA, long-chain poly-unsaturated fatty acids). Log_2_ values were conditionally formatted within each metabolite across diet on a color scale. Blue indicates a relative concentration lower than the other treatments, white indicates an intermediate relative concentration, and red indicates a higher relative concentration than the other treatments. 1, heptanoate (7:0); 2, caprylate (8:0); 3, caprate (10:0); 4, laurate (12:0); 5, 5-dodecenoate (12:1n7); 6, myristate (14:0); 7, pentadecanoate (15:0); 8, palmitate (16:0); 9, margarate (17:0); 10, stearate (18:0); 11, nonadecanoate (19:0); 12, arachidate (20:0); 13, palmitoleate (16:1n7); 14, 10-heptadecenoate (17:1n7); 15, oleate/vaccenate (18:1); 16, 10-nonadecenoate (19:1n9); 17, erucate (22:1n9); 18, hexadecadienoate (16:2n6); 19, hexadecatrienoate (16:3n3); 20, linoleate (18:2n6); 21, stearidonate (18:4n3); 22, arachidonate (20:4n6); 23, eicosapentaenoate (EPA; 20:5n3); 24, heneicosapentaenoate (21:5n3); 25, docosadienoate (22:2n6); 26, docosatrienoate (22:3n3); 27, docosapentaenoate (n6 DPA; 22:5n6); 28, docosahexaenoate (DHA; 22:6n3).

**Table 1 microorganisms-09-02293-t001:** Estimated mean (and standard error) total tract digestibility, food intakes and fecal output of dogs (*n* = 6 dogs per treatment) subjected to a digestibility study involving high, medium and low shear dietary treatments (HS, MS and LS, respectively).

Item	HS	MS	LS
ATTD, %			
Dry Matter	84.6 ± 0.67	85.5 ± 0.42	85.9 ± 0.44
Organic Matter	88.4 ± 0.42	89 ± 0.31	88.7 ± 0.33
Gross Energy	88.3 ± 0.46	89.3 ± 0.18	88.9 ± 0.29
Crude Protein	84.1 ^b^ ± 0.63	84.6 ^b^ ± 0.55	86.7 ^a^ ± 0.38
True Protein	91.4 ± 1.07	91.6 ± 0.60	92.9 ± 0.43
Crude Fat	93.9 ± 0.26	93.5 ± 0.35	94.3 ± 0.46
Ash	25.0 ^b^ ± 2.81	27.0 ^b^ ± 2.81	43.9 ^a^ ± 2.98
Responses			
Daily Intake, g·day^−1^ (as-is)	149 ± 15.0	169 ± 16.4	165 ± 15.3
Daily Intake, g·day^−1^ (DM)	136 ± 13.7	152 ± 14.8	148 ± 13.7
Daily Intake, Kcal·day^−1^	556 ± 56.0	628 ± 61.2	615 ± 57.0
Fecal Output, g·day^−1^ (as-is)	64.3 ± 6.83	70.9 ± 7.29	66 ± 6.29
Fecal Output, g·day^−1^ (DM)	21.1 ± 2.67	21.9 ± 2.00	20.7 ± 1.49

^a,b^ Letters indicate significant pairwise differences (*p* < 0.05).

**Table 2 microorganisms-09-02293-t002:** Food intake and fecal parameters (estimated least squared means and [95% confidence interval]) of dogs fed dietary treatments produced at high, medium, and low shear (HS, MS and LS, respectively).

Item	HS	MS	LS	*p* (F)
Intake				
Food, g/BW^0.75^/day	33.2 [30.0, 36.7]	33.1 [30.0, 36.6]	31.8 [28.8, 35.1]	0.0317
Food, Kcal/BW^0.75^/day	154.5 ^a^ [139.7, 170.7]	153.5 ^a^ [138.8, 169.7]	144.4 ^b^ [130.6, 159.7]	0.0007
Resistant Starch, g/BW^0.75^/day	0.206 ^c^ [0.189, 0.224]	0.295 ^b^ [0.273, 0.318]	0.319 ^a^ [0.297, 0.342]	<0.0001
Fecal Responses				
Moisture, %	69.7 [68.5, 70.9]	69.8 [68.6, 71.0]	68.7 [67.5, 69.9]	0.2200
Organic Matter, %	22.4 [21.8, 23.0]	22.7 [22.1, 23.3]	23.1 [22.4, 23.7]	0.2522
pH	5.82 [5.70, 5.93]	5.72 [5.61, 5.84]	5.67 [5.56, 5.78]	0.0719
Ammonium, mmol/g	0.034 [0.032, 0.037]	0.032 [0.028, 0.035]	0.031 [0.026, 0.035]	0.1987
Ash, %	7.66 [6.85, 8.56]	7.09 [6.34, 7.93]	8.15 [7.29, 9.11]	0.0986

^a,b,c^ Letters indicate significant pairwise differences (*p* < 0.05).

**Table 3 microorganisms-09-02293-t003:** Estimated mean concentration (and [95% confidence interval]) of fecal short-chain fatty acids (SCFA), branched-chain fatty acids (BCFA) and immunoglobulin a, as well as serum gram-positive and gram-negative bacterial cell wall constituents lipoteichoic acid (LTA) and lipopolysaccharide (LPS), respectively, of fasting dogs fed dietary treatments s produced at high, medium and low shear (HS, MS and LS, respectively).

Item			HS	MS	LS	*p* (F)
Feces						
	SCFA, mg/g					
		Total	10.41 [9.26, 11.56]	10.20 [9.06, 11.35]	11.01 [9.86, 12.16]	0.1700
		Acetic acid	4.73 [4.31, 5.14]	4.26 [3.87, 4.66]	4.59 [4.03, 5.15]	0.1243
		Propionic acid	3.23 [2.80, 3.66]	2.77 [2.35, 3.20]	3.10 [2.67, 3.52]	0.1572
		Butyric acid	2.19 ^b^ [1.77, 2.72]	2.68 ^ab^ [2.16, 3.32]	3.01 ^a^ [2.43, 3.73]	0.0236
	BCFA, mg/g					
		Total	0.582 [0.386, 0.878]	0.667 [0.437, 1.019]	0.654 [0.406, 1.055]	0.5992
		2-Methylbutyric acid	0.108 [0.081, 0.134]	0.098 [0.071, 0.125]	0.103 [0.074, 0.133]	0.4744
		Isobutyric acid	0.159 [0.130, 0.188]	0.144 [0.115, 0.172]	0.145 [0.116, 0.173]	0.5150
		Isovaleric Acid	0.200 [0.163, 0.238]	0.186 [0.148, 0.224]	0.190 [0.140, 0.240]	0.6484
	IgA, mg/g		19.2 [11.0, 33.5]	23.7 [13.6, 41.3]	20.0 [11.5, 34.9]	0.6174
Serum						
	LTA, ng/mL		4.96 [3.75, 6.17]	5.05 [3.84, 6.26]	5.27 [4.06, 6.48]	0.8384
	LPS, Eu/mL		1.18 [0.95, 1.40]	1.35 [1.12, 1.57]	1.38 [1.15, 1.60]	0.0763

^a,b^ Letters indicate significant pairwise differences (*p* < 0.05).

**Table 4 microorganisms-09-02293-t004:** Estimated mean concentration (and [95% confidence interval]) of satiety hormones in plasma of fasting dogs (*n* = 23) fed diets produced at high, medium, and low shear (HS, MS and LS, respectively).

Satiety Hormone	HS	MS	LS	*p* (F)
Ghrelin, pg/mL	105.3 [79.6, 139.4]	118.8 [89.9, 157.0]	112.7 [85.2, 149.1]	0.3853
Leptin, pg/mL	252.3 [159.2, 366.8]	312.1 [207.8, 437.6]	277.3 [179.5, 396.3]	0.1379
GIP, pg/mL	3.05 [2.30, 4.05]	2.49 [1.88, 3.29]	2.81 [2.12, 3.72]	0.1968
Glucagon, pg/mL	41.8 [29.0, 60.1]	42.5 [29.0, 62.3]	41.2 [28.2, 60.3]	0.7462
Insulin, pg/mL	41.6 [26.4, 65.7]	51.4 [32.7, 81.0]	61.1 [38.7, 96.3]	0.0648
PP, pg/mL	53.2 [41.2, 68.8]	53.1 [41.3, 68.3]	57.2 [44.4, 73.7]	0.8401
PYY, pg/mL	101.5 [89.8, 114.7]	106.9 [94.7, 120.7]	106.4 [94.2, 120.1]	0.4533

**Table 5 microorganisms-09-02293-t005:** Estimated mean (and [95% confidence interval]) alpha diversity indices for richness, inverse simpson, exponent shannon and Pileou’s evenness (J) of the microbiota of dogs (*n* = 24) fed diets produced at high, medium and low shear (HS, MS and LS, respectively).

Indices	HS	MS	LS	*p*
Richness	75.0 [71.9, 78.0]	76.1 [73.0, 79.1]	76.6 [73.5, 79.6]	0.7438
InvSimp	11.7 [10.0, 13.4]	11.9 [10.6, 13.3]	11.9 [10.2, 13.6]	0.9581
ExpH	18.4 [16.7, 19.9]	18.6 [17.0, 20.1]	18.4 [16.7, 19.9]	0.9644
Pileou’s J	0.667 [0.645, 0.688]	0.666 [0.645, 0.685]	0.658 [0.629, 0.684]	0.7726

**Table 6 microorganisms-09-02293-t006:** Estimated means Operational Taxonomic Unit (OTU) (±Standard Error, expressed in the Centered Log-Ratio scale) of dogs fed diets produced at high, medium and low shear (HS, MS and LS, respectively).

Phylum	OTU ^1^	HS	MS	LS	*p*	q
Firmicutes	361186 Lachnospiraceae Blautia	5.25 ^a^ ± 0.148	4.89 ^b^ ± 0.148	5.05 ^ab^ ± 0.148	0.0618	0.2645
	99508 Turicibacteraceae Turicibacter	4.23 ^a^ ± 0.171	3.72 ^b^ ± 0.169	4.01 ^ab^ ± 0.169	0.0266	0.1966
	100212 Veillonellaceae	0.88 ^a^ ± 0.258	0.18 ^b^ ± 0.257	0.35 ^ab^ ± 0.263	0.0373	0.1966
	28914 Lachnospiraceae Roseburia	−0.59 ^b^ ± 0.260	0.18 ^a^ ± 0.260	−0.32 ^ab^ ± 0.260	0.0377	0.1966
	40839 Erysipelotrichaceae Catenibacterium	−0.81 ^b^ ± 0.214	−0.19 ^a^ ± 0.214	−0.58 ^ab^ ± 0.214	0.0305	0.1966
Actinobacteria	367139 Coriobacteriaceae Slackia	3.20 ^a^ ± 0.229	2.40 ^b^ ± 0.234	2.72 ^ab^ ± 0.228	0.0070	0.1580
Bacteroidetes	102407 Bacteroidaceae Bacteroides	0.58 ^b^ ± 0.396	1.75 ^a^ ± 0.396	0.92 ^ab^ ± 0.396	0.0117	0.1584
	1000062 Porphyromonadaceae unclassified	1.23 ^a^ ± 0.197	0.79 ^b^ ± 0.200	0.47 ^b^ ± 0.200	0.0004	0.0272
	1105615 Paraprevotellaceae	−2.31 ± 0.513	−1.33 ± 0.513	−2.29 ± 0.513	0.0555	0.2645
	1066621 Prevotellaceae unclassified	−2.42 ^b^ ± 0.469	−1.31 ^a^ ± 0.469	−2.38 ^b^ ± 0.469	0.0219	0.1966
Fusobacteria	298592 Fusobacteriaceae	1.24 ^a^ ± 0.214	0.46 ^b^ ± 0.211	0.99 ^ab^ ± 0.214	0.0039	0.1337
	838467 Fusobacteriaceae Cetobacterium	0.68 ^a^ ± 0.225	0.00 ^b^ ± 0.225	0.42 ^ab^ ± 0.225	0.0335	0.1966
Proteobacteria	4319416 Bartonellaceae Bartonella	0.733 ^a^ ± 0.24	0.085 ^b^ ± 0.24	0.241 ^ab^ ± 0.24	0.0110	0.1584
	10001 Enterobacteriaceae unclassified	−1.41 ^a^ ± 0.141	−1.87 ^b^ ± 0.141	−1.56 ^ab^ ± 0.141	0.0264	0.1966
	527323 Enterobacteriaceae Yersinia	−3.38 ^b^ ± 0.192	−3.28 ^ab^ ± 0.192	−2.79 ^a^ ± 0.192	0.0254	0.1966

^1^ OTU expressed at the Family_Genera level, ordered by decreasing abundance within each Family. ^a,b^ Letters indicate significant pairwise differences *p* < 0.05.

**Table 7 microorganisms-09-02293-t007:** Estimated Spearman rho’s correlations between selected operational taxonomic units (OTUs; expressed in centered log-ratio transformation) and concentration of fecal SCFA, lactate, pH, oligosaccharides and succinate (expressed as log_2_).

	Fecal Short-Chain Fatty Acids				Fecal Sugars			Fecal Fermentation Markers		
OTU^1^	Total SCFA	Butyric Acid	Acetic Acid	Propionic Acid	Glucose	Maltose	Malto-Tetraose	Lactate	pH	Succinate
Lachnospiraceae Blautia	−0.52 ^a^	−0.51 ^a^	0.05	−0.07	0.06	−0.32	−0.18	0.02	0.25	−0.19
Turicibacteraceae Turicibacter	−0.35	−0.26	−0.02	−0.08	0.09	−0.18	−0.08	0.06	0.16	−0.07
Veillonellaceae	0.01	−0.08	0.36	0.09	−0.13	−0.08	−0.25	−0.13	0.2	−0.27
Lachnospiraceae Roseburia	0.27	0.17	−0.08	0.12	−0.01	0.35	0.19	0.04	−0.21	0.11
Erysipelotrichaceae Catenibacterium	−0.03	−0.19	0.03	0.06	0.38 ^b^	0.21	0.17	0.00	−0.15	−0.04
Coriobacteriaceae Slackia	−0.47 ^a^	−0.37	−0.06	−0.10	0.17	−0.23	0.04	0.09	0.13	−0.18
Bacteroidaceae Bacteroides	0.38	0.67 ^a^	−0.26	−0.20	−0.30	0.14	−0.07	−0.13	0.06	0.16
Porphyromonadaceae unclassified	−0.06	−0.32	0.23	0.27	0.14	−0.05	−0.07	−0.12	−0.01	−0.34
Paraprevotellaceae	0.37	0.55 ^a^	−0.12	−0.18	−0.36	0.01	−0.21	−0.21	0.00	0.16
Prevotellaceae unclassified	0.36	0.59 ^a^	−0.18	−0.18	−0.37	0.08	−0.16	−0.14	−0.01	0.25
Fusobacteriaceae	−0.30	−0.33	0.06	0.08	0.20	−0.17	0.07	0.16	0.08	−0.11
Fusobacteriaceae Cetobacterium	−0.09	−0.05	0.12	0.06	0.04	−0.11	−0.13	−0.24	0.40 ^c^	−0.46 ^c^
Bartonellaceae Bartonella	0.17	−0.11	0.41 ^a^	0.25	−0.12	−0.24	−0.34	−0.19	0.09	−0.35
Enterobacteriaceae unclassified	−0.25	−0.31	0.08	0.05	−0.03	−0.30	−0.19	−0.06	0.30	−0.25
Enterobacteriaceae Yersinia	0.03	−0.03	−0.06	−0.05	0.16	0.08	0.27	0.29	−0.36	0.34

^a^ Correlation coefficients for fecal SCFA which are significant with *p* < 0.00084 (adjusted *p* < 0.05). ^b^ Correlation coefficients for fecal sugars which are significant with *p* < 0.00111 (adjusted *p* < 0.05). ^c^ Correlation coefficients for fecal fermentation markers which are significant with *p* < 0.00111 (adjusted *p* < 0.05).

## Data Availability

Data included in the manuscript or available from the corresponding author.

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
