# Peer review of "A Low to Medium-Shear Extruded Kibble with Greater Resistant Starch Increased Fecal Oligosaccharides, Butyric Acid, and Other Saccharolytic Fermentation By-Products in Dogs"

_microorganisms, 2021, doi:10.3390/microorganisms9112293_

Round 1

Reviewer 1 Report

The study " A low to medium-shear extruded kibble with greater resistant starch increased fecal oligosaccharides, butyrate, and other saccharolytic fermentation by-products in dogs" tested the effect of different amounts of resistant starch as generated by extrusion at different levels of mechanical force. The study was very well done and clearly presented. A strength of the study is the fact that a initial pilot study was performed to confirm that there were no changes in dietary behavior with the different extruded diets. The introduction clearly give justification for the experiment and the methods are appropriately described and conclusions are consistent with results. 

The are only the following minor edits:  

Line 15: change "on fresh" to from fresh

Line 25-25: Change to "innovation for additional health benefits for their commercial"

Line 148: ground up

Line 206: "measured in dog"

Table 1: Make sure LS in the heading is at the same level as HS and MS

 Line 420: "in fecal metabolites"

Line 422: "while dogs fed the"

Line 487: Correct Figure 3 legend (not figure 1)

Line 506" Correct Figure 4 legend (not figure 2)

Line 613: Define SDS the first time you use it. 

Line 636-639: Reword sentence.  

Line 742: Correct homeostasis

Author Response

The study " A low to medium-shear extruded kibble with greater resistant starch increased fecal oligosaccharides, butyrate, and other saccharolytic fermentation by-products in dogs" tested the effect of different amounts of resistant starch as generated by extrusion at different levels of mechanical force. The study was very well done and clearly presented. A strength of the study is the fact that a initial pilot study was performed to confirm that there were no changes in dietary behavior with the different extruded diets. The introduction clearly give justification for the experiment and the methods are appropriately described and conclusions are consistent with results. 

Thank you for your note, and for the time you dedicated into reviewing our work. Please see replies to specific comments below.

The are only the following minor edits:  

Line 15: change "on fresh" to from fresh Change made. Thank you.

Line 25-25: Change to "innovation for additional health benefits for their commercial" Change made. Thank you.

Line 148: ground up. Thanks, but just “ground” would be the most formal and grammatically correct.

Line 206: "measured in dog" Change made. Thank you.

Table 1: Make sure LS in the heading is at the same level as HS and MS We center aligned the three treatments. It doesn’t show on track changes, but it is correct when you select “No markup”. Thanks.

 Line 420: "in fecal metabolites" Change made. Thank you.

Line 422: "while dogs fed the" Change made. Thank you.

Line 487: Correct Figure 3 legend (not figure 1) Thank you for noticing it, it has been corrected.

Line 506" Correct Figure 4 legend (not figure 2) Change made. Thank you.

Line 613: Define SDS the first time you use it. We defined it. Thank you.

Line 636-639: Reword sentence.  We agree this phrase was awkward. It was modified to: “The small intestine (SI) is sensitive to dietary changes and has a higher abundance of carbohydrate fermenting bacteria than the colon [27, 28]. Thus, considering that feces of dogs fed the LS or MS diets had more oligosaccharides than those fed the HS it would suggest the presence of higher concentrations of these sugars in both the SI and proximal regions of the large intestine.”

Line 742: Correct homeostasis Corrected. Thank you.

Submission Date

05 October 2021

Date of this review

14 Oct 2021 21:16:57

Reviewer 2 Report

General Remarks

This article reports work assessing how increasing dietary Resistant Starch affects markers of colonic health in dogs. It has involved a huge amount of work, with many analyses covering many different aspects of digestion and fermentation in the dog gut. Once concern I have, is that there almost seems to be too much, and the reader becomes lost in all the incredible detail. I missed an overarching idea or to explain the findings, which surely hinge on differences in RS between diets? In the end, the final conclusion is that increased RS may be beneficial for dog health due to a small increase in butyrate. I’m sure there could be so much more that you could get out of the incredible wealth of information you have here.

For example, it is interesting that the ash digestibility for LS is almost twice that of HS and MS? The observation is made, but there doesn’t seem to be any suggestion as to why this could be the case - this could be explored further.

Please beware the use of terminology “microbiota” versus “microbiome”. Sometimes they seem to be used almost interchangeably, but of course “biota” refers to actual microbes, and “biome” to their genes. Please examine the entire text and correct appropriately.

There are further concerns which have been addressed below.

(I am not qualified to be critical of the methodology used for the “Gut Health Study”)

Detailed Remarks

L26         …for their commercial products.

L27         …formulae…

L28         What is meant by “value-based” ingredients? Is this qualification necessary?

L157       Given that the dogs had previously been fed the day before at 7.30am, I’m wondering why blood samples were collected in a fasting state (~ 24hours later) to measure satiety hormones. Do you expect to see many differences between diets so long after the last meal?

L375       …nor on fecal ash….

L540       What is meant by “Scheme 0.”?

L541       Presumably here you mean “…were not statistically significantly different…”…. (One could leave out “statistically” for the sake of brevity.

L544       “…for a statistical significance”? Significant what? Increase? Decrease? Difference? Perhaps it would be more accurate to say something like “…whereas dogs fed LS showed no evidence of being significantly different compared with MS and HS….”

L548       This is not Table 1- Table 7?

L550       It is not completely clear what “a, b, and c”, stand for if one looks at the actual table. It might be clearer to say something like “Correlation coefficients for fecal SCFA which are significant with P<0.0008 (adjusted P<0.05)” if this is actually the case. The implication is that the other values are not significant? Is that correct?

L559       …was above the average….

L562       …correlation with….

L565       … of a dietary effect…

L571       Please be consistent throughout the paper- either acetic, propionic and butyric acids, or acetate, propionate and butyrate, but not used interchangeably.

L579-8   Not sure of the wording of this sentence. Isn’t RS itself a potentially beneficial energy substrate, rather than being turned into one? Please clarify.

L598       Presumably your work is pointing to a change in microbial activity, instead of the bacteria which are present. You might like to mention this, as like this it is left a bit open-ended.

L602       …such as… (rather than “like”)

L602       …. , , species mastication habits, … is there a factor missing between the two commas?

L619-20                 …thus lower starch cook than…. What is meant by this? Do you mean that “thus, starch was cooked to a lesser extent….?

L621       …with regard to changes in the …..

L627       …the increased faecal butyrate, when dogs were fed the LS relative to HS implies an improvement in gut health, which is typical of that of a prebiotic…. I’m not sure that this is strictly accurately phrases. Increased faecal butyrate does not imply an improvement in gut health as such, as it is not a measure of this. For example, a positive change in a Comet test could imply such an improvement. What increased Butyrate does do, is imply a “potential” improvement in gut health, which could be tested by…. (a range of physiological tests which you could insert here, depending on what you may like to do in the future?).

L642       …microbiota….

L660       Just for readability, would it be better to say “…dogs fed more RS (i.e. MS and LS) tended to have lower fecal pH….

L662-4   In this sentence the use of verbs (…was the genera…) suggests that in this study Roseburia had the most saccharolytic activity. But then in L 665, it is said that “in this study”, that is not the case? Or, is there no link between saccharolytic activity and production of butyrate and SCFA? I find this section rather confusing… Please rewrite for clarity.

L671       ….of human studies…

L681-2   …had a tendency to decrease in dogs fed the LS and MS foods…. (not in the foods themselves)

L699       Why not just “health” rather than healthfulness….?

L724       …such as glucose….

L728       …linked to the attempt of bacteria to decrease reducing agents in their environments… Please expand on this- which reducing agents do you have in mind?

L737       How about… …has been a preferred end-product for studies of prebiotics….?

L739      This is not strictly accurate- SCFA are not the preferred energy sources for colonocytes, only butyrate, as far as I know. Having said that (and as the authors go on to say here)- acetate and propionate have many other functions in the body, as they are both absorbed into the circulatory system. I think it would be better in this paragraph to be more specific about individual actions of each SCFA rather than just lumping them all together.  

L747       This sentence is very confusing. ….In the present study, there ….

L748       …propionate concentrations….

L762       Please indicate how it is known in this case, that these changes are “directly” linked to starch fermentation.

L764-5   Please bear in mind that luminal pH is not only reflection of SCFA presence, but also of SCFA absorption and bicarbonate secretion into the lumen, amongst other factors (as you mention later). Were SCFA in serum measured to give an indication of tendencies for differences between diets for SCFA absorbed?

L769       Actually, whether the pH would be lower in the colon or faeces depends on what is in the diet and where it is fermented. This would be a reasonable assumption if there were only rapidly fermentable ingredients in the diet, but I’m not sure if this would be the case with the diets being examined here. Has any work been done to look at a measure of fermentabilty of these diets- most likely in vitro? I think this statement will need to be more nuanced….

L776       …distal… (not distant)

L790       If you didn’t expect any changes in fasting dogs, why did you measure these serum parameters. They don’t seem to add anything to your narrative.

L805-7   If the HS diet dogs had more fecal LCPUFA, but not serum values, what is your explanation for this? An example of bacteria which could be involved are mentioned, but this is not then tied together with what was actually found.

L808       Saccharolytic bacteria … such as….

L815       …ileum…

Table S5

Given that different letters indicate treatment differences with a P value of <0.05, how can the values for Calcium be significantly different (between MS and LS) when the recorded P value is 0.062?

Table S6

How can values for amino acid concentrations be negative? I realize that these are mean values, but shouldn’t they then be “0”?  It might help to show the n value in this table, and then the maximum and minimum values (as for Table….)

Author Response

Thank you for the time and effort you dedicated into reviewing our work. Your comments and additions were very much appreciated by the authors. We agree there is a lot of information, but all results presented were related to one another and helped us strengthen our hypothesis. We agree microbiota and microbiome were used interchangeably. We changed this term all to “microbiota” because we didn’t measure the genes and were referring to the microbial population. Also, we added in the discussion about ash digestibility: “Ash digestibility on the preliminary assessment was greater when dogs were fed the diet containing more RS (LS food), which corroborates findings from other studies [27, 28] regarding an increased absorption of some minerals when prebiotics were added to the dog and human foods.”

Detailed Remarks

L26         …for their commercial products. Thank you. This was also a correction from reviewer 1 and was made.

L27         …formulae… Change made. Thank you.

L28         What is meant by “value-based” ingredients? Is this qualification necessary? We changed it for “commonly used”. Thanks.

L157       Given that the dogs had previously been fed the day before at 7.30am, I’m wondering why blood samples were collected in a fasting state (~ 24hours later) to measure satiety hormones. Do you expect to see many differences between diets so long after the last meal?

There is research that points towards a higher concentration in satiety/hunger hormones PYY and GLP-1 when the subjects have increased acetate and propionate in the circulation. So, we wouldn’t expect changes in SCFA to occur a few hours after a meal, but as a delayed response. We added this information on the discussion: “All SCFAs; butyrate, propionate and acetate, have been reported to regulate satiety in the long-term via activation of PYY and GLP-1 and increased expression of leptin, exerting an anorexigenic effect [56].”  and concluded “It is possible that there wasn’t enough RS to promote satiety through SCFA production, and(or) that dogs were fasting for a longer time (> 20 h) which overrode changes in satiety hormones.”

L375       …nor on fecal ash…. Change made. Thank you.

L540       What is meant by “Scheme 0.”? Thank you for catching this mistake. The beginning of that paragraph was cut when formatting the manuscript into the Microorganisms template. We added back this section: “Some of the most abundant OTUs had no evidence of any dietary effects, including Prevotellaceae Prevotella, Clos-tridiaceae Clostridium, Lactobacillaceae Lactobacillus, Alcaligenaceae Sutterella, Lachnospiraceae Ruminococcus and Fusobacteriaceae Fusobacterium. In contrast, the most abundant OTUs for which significant treatment effects were identified, were within Phylum Firmicutes. Specifically, Lachnospiraceae Blautia was the most abundant OTU with a marginal significance (P = 0.06; q > 0.10);”

L541       Presumably here you mean “…were not statistically significantly different…”…. (One could leave out “statistically” for the sake of brevity. We agree. That was changed to “was not different”

L544       “…for a statistical significance”? Significant what? Increase? Decrease? Difference? Perhaps it would be more accurate to say something like “…whereas dogs fed LS showed no evidence of being significantly different compared with MS and HS….” Thank you for your comment. We simplified that to “, whereas dogs fed LS were not different from MS and HS.”

L548       This is not Table 1- Table 7? Thank you. We just corrected this mistake; it also happened when reformatting the ms.

L550       It is not completely clear what “a, b, and c”, stand for if one looks at the actual table. It might be clearer to say something like “Correlation coefficients for fecal SCFA which are significant with P<0.0008 (adjusted P<0.05)” if this is actually the case. The implication is that the other values are not significant? Is that correct? Yes, this is correct. Suggested changes were made to increase clarity.

L559       …was above the average…. Change made. Thank you.

L562       …correlation with…. Change made. Thank you.

L565       … of a dietary effect… Change made. Thank you.

L571       Please be consistent throughout the paper- either acetic, propionic and butyric acids, or acetate, propionate and butyrate, but not used interchangeably. Thank you, we left them all in the acid form.

L579-8   Not sure of the wording of this sentence. Isn’t RS itself a potentially beneficial energy substrate, rather than being turned into one? Please clarify. The energy substrates that we refer in this sentence are the SCFA. We modified it to “…and reaches the large intestine where it is fermented by commensal bacteria into beneficial energy substrates like SCFAs [21]”

L598       Presumably your work is pointing to a change in microbial activity, instead of the bacteria which are present. You might like to mention this, as like this it is left a bit open-ended. Great comment. We added a conclusion phrase to this paragraph: “Thus, data from the present work point towards a change in microbial activity without significantly changing the microbial ecology.”

L602       …such as… (rather than “like”) Changed.

L602       …. , , species mastication habits, … is there a factor missing between the two commas? This was a typo and was corrected. Thank you.

L619-20                 …thus lower starch cook than…. What is meant by this? Do you mean that “thus, starch was cooked to a lesser extent….? We understand this phrase can be confusing, so it was modified to: “retained more RS and SDS, because starch was cooked to a lesser extent than the other diets [7]”. Initially, we were referring to the starch cook enzymatic procedure, but it is clearer to the reader after the modification. Thank you.

L621       …with regard to changes in the ….. Change made. Thank you.

L627       …the increased faecal butyrate, when dogs were fed the LS relative to HS implies an improvement in gut health, which is typical of that of a prebiotic…. I’m not sure that this is strictly accurately phrases. Increased faecal butyrate does not imply an improvement in gut health as such, as it is not a measure of this. For example, a positive change in a Comet test could imply such an improvement. What increased Butyrate does do, is imply a “potential” improvement in gut health, which could be tested by…. (a range of physiological tests which you could insert here, depending on what you may like to do in the future?). Thank you. We added the word “potential” before improvement. We did not add potential benefits from butyrate, because it is described in another section of the discussion.

L642       …microbiota…. Changed. Thank you.

L660       Just for readability, would it be better to say “…dogs fed more RS (i.e. MS and LS) tended to have lower fecal pH…. Agreed, thanks.

L662-4   In this sentence the use of verbs (…was the genera…) suggests that in this study Roseburia had the most saccharolytic activity. But then in L 665, it is said that “in this study”, that is not the case? Or, is there no link between saccharolytic activity and production of butyrate and SCFA? I find this section rather confusing… Please rewrite for clarity. Thank you for noticing that. We removed the first part and reworded this sentence for clarity: “Anaerobic gram-positive Roseburia within the phylum Firmicutes has the ability to utilize starch and produce butyrate [21, 35]. However, findings from this study indicate that this microbe was not correlated to butyrate or SCFA.”

L671       ….of human studies… Corrected. Thank you.

L681-2   …had a tendency to decrease in dogs fed the LS and MS foods…. (not in the foods themselves) True, thanks for catching that.

L699       Why not just “health” rather than healthfulness….? We changed it to “health”. Thanks.

L724       …such as glucose….Changed, thanks.

L728       …linked to the attempt of bacteria to decrease reducing agents in their environments… Please expand on this- which reducing agents do you have in mind? We added: “…decrease reducing agents equivalents (such as 2H+ and NADH) in their environment.” Thank you.

L737       How about… …has been a preferred end-product for studies of prebiotics….? Change made. Thanks.

L739      This is not strictly accurate- SCFA are not the preferred energy sources for colonocytes, only butyrate, as far as I know. Having said that (and as the authors go on to say here)- acetate and propionate have many other functions in the body, as they are both absorbed into the circulatory system. I think it would be better in this paragraph to be more specific about individual actions of each SCFA rather than just lumping them all together.  Thank you. We modified and simplified the first part of this paragraph: “Butyric acid is mainly used as colonocyte energetic substrate, but also presents anti-inflammatory properties that improve intestinal homeostasis and mucosa immunity [32, 56]. All SCFAs; butyric, propionic and acetic acids, have been reported to regulate satiety in the long-term via activation of PYY and GLP-1 and increased expression of leptin, exerting an anorexigenic effect [56]. These satiety effects may be attributed mainly to propionic and acetic acids, since butyric acid is mostly utilized at the intestinal level [57].”

L747       This sentence is very confusing. ….In the present study, there …. Thank you, we reformulated this sentence: “In the present study there was no evidence for a dietary treatment effect on propionic and acetic acids concentrations, nor on satiety in the long term.

L748       …propionate concentrations….Change made. Thanks.

L762       Please indicate how it is known in this case, that these changes are “directly” linked to starch fermentation. The degree of cooking was the only difference among diets, which were extruded from the same batch. So, it is very likely that these differences are due to different levels of resistant starch. We have revised this phrase: “Dogs fed the LS or MS diets showed marginal evidence for an increase in lactate, succinate and butyric acid, and decreased fecal pH, which were indicative of starch fermentation by saccharolytic bacteria [53, 58].”

L764-5   Please bear in mind that luminal pH is not only reflection of SCFA presence, but also of SCFA absorption and bicarbonate secretion into the lumen, amongst other factors (as you mention later). Were SCFA in serum measured to give an indication of tendencies for differences between diets for SCFA absorbed? Serum metabolomics was performed and there were some molecules which could be derivatives of SCFAs. This data was not presented because there were no differences in serum SCFA derivatives after the q-values were calculated.

L769       Actually, whether the pH would be lower in the colon or faeces depends on what is in the diet and where it is fermented. This would be a reasonable assumption if there were only rapidly fermentable ingredients in the diet, but I’m not sure if this would be the case with the diets being examined here. Has any work been done to look at a measure of fermentabilty of these diets- most likely in vitro? I think this statement will need to be more nuanced…. We haven’t conducted an “in vitro” evaluation of these diets. The idea that pH would be higher in the feces is because the SCFA would have had more time to be absorbed with an exchange of bicarbonate by the colonocytes, that would increase pH in the rectum or feces. This finding was reported by den Besten (2013). We revised our sentence to be more nuanced: “So, it is possible that dog colonic pH would be lower than what was measured in the rectum or feces.”

L776       …distal… (not distant) Thank you for catching it (this typo made me laugh).

L790       If you didn’t expect any changes in fasting dogs, why did you measure these serum parameters. They don’t seem to add anything to your narrative. We did not expect changes in serum metabolites, but wanted to confirm that this was actually the case. We added at the end of the last sentence: “.., and this was confirmed.”

L805-7   If the HS diet dogs had more fecal LCPUFA, but not serum values, what is your explanation for this? An example of bacteria which could be involved are mentioned, but this is not then tied together with what was actually found. Thank you for pointing this out. To address the serum discrepancy: Since fatty acids longer than about 6 carbons are not readily absorbed across the colon in a quantitatively meaningful sense we would not expect there to be equilibration of the serum levels of fatty acids with fecal fatty acids of the chain lengths we are describing. We added this information on now L813-815: “This would be expected since long-chain fatty acids are not readily absorbed across the colon in a quantitatively meaningful manner, so serum LCPUFAs should not be directly correlated with its amount in feces.” We also added another conclusion about lipid metabolism: “The tendency to decrease fecal medium chain fatty acids (MCFAs), as well as some mono- and diacylglycerols when dogs were fed the HS food may suggest that the restriction of fermentable carbohydrate induced microbes to seek alternative carbon sources.”

L808       Saccharolytic bacteria … such as…. Change made. Thanks.

L815       …ileum… Change made. Thanks.

Table S5

Given that different letters indicate treatment differences with a P value of <0.05, how can the values for Calcium be significantly different (between MS and LS) when the recorded P value is 0.062? We have removed the superscripts for this trend. Thank you.

Table S6

How can values for amino acid concentrations be negative? I realize that these are mean values, but shouldn’t they then be “0”?  It might help to show the n value in this table, and then the maximum and minimum values (as for Table….) Thank you for your question. We understand this can be a bit confusing. The negative numbers are due to centered log-ratio transformation. Numbers that are exactly 0 are at the mean, numbers below 0 are lower than the mean, and numbers above zero are above the mean. Please see L288-299:

“zero OTU counts were inflated using the Bayesian multiplicative treatment [16] and converted to centered log-ratio (CLR) to normalize the arbitrary counts determined by the DNA sequencer:

Where OTUX = counts of one OTU in the dataset; GOTU = geometric mean of all OTU counts measured within a fecal sample. The CLR transformation converts each OTU count into a ratio relative to the geometric mean of all OTUs within each fecal sample [17]. Some properties of CLR are that the sum of all OTUs within a sample converge to zero (the mean), the reads are normalized to a common sequencing depth, and the CLR transformed values are scale invariant. This means that the same ratio is expected to be obtained in a sample with few read counts as a cohort sample with many read counts [17].”
